# Pseudogene Transcripts in Head and Neck Cancer: Literature Review and *In Silico* Analysis

**DOI:** 10.3390/genes12081254

**Published:** 2021-08-17

**Authors:** Juliana Carron, Rafael Della Coletta, Gustavo Jacob Lourenço

**Affiliations:** 1Laboratory of Cancer Genetics, School of Medical Sciences, University of Campinas, Campinas 13083-888, São Paulo, Brazil; julianacarron@outlook.com.br; 2Department of Agronomy and Plant Genetics, University of Minnesota, Saint Paul, MN 55108, USA; della028@umn.edu

**Keywords:** head and neck cancer, pseudogene transcripts, SNV, co-expression network, gene ontology enrichment

## Abstract

Once considered nonfunctional, pseudogene transcripts are now known to provide valuable information for cancer susceptibility, including head and neck cancer (HNC), a serious health problem worldwide, with about 50% unimproved overall survival over the last decades. The present review focuses on the role of pseudogene transcripts involved in HNC risk and prognosis. We combined current literature and *in silico* analyses from The Cancer Genome Atlas (TCGA) database to identify the most deregulated pseudogene transcripts in HNC and their genetic variations. We then built a co-expression network and performed gene ontology enrichment analysis to better understand the pseudogenes’ interactions and pathways in HNC. In the literature, few pseudogenes have been studied in HNC. Our *in silico* analysis identified 370 pseudogene transcripts associated with HNC, where *SPATA31D5P*, *HERC2P3*, *SPATA31C2*, *MAGEB6P1*, *SLC25A51P1*, *BAGE2*, *DNM1P47*, *SPATA31C1*, *ZNF733P* and *OR2W5* were found to be the most deregulated and presented several genetic alterations. *NBPF25P*, *HSP90AB2P*, *ZNF658B* and *DPY19L2P3* pseudogenes were predicted to interact with 12 genes known to participate in HNC, *DNM1P47* was predicted to interact with the *TP53* gene, and *HLA-H* pseudogene was predicted to interact with *HLA-A* and *HLA-B* genes. The identified pseudogenes were associated with cancer biology pathways involving cell communication, response to stress, cell death, regulation of the immune system, regulation of gene expression, and Wnt signaling. Finally, we assessed the prognostic values of the pseudogenes with the Kaplan–Meier Plotter database, and found that expression of *SPATA31D5P*, *SPATA31C2*, *BAGE2*, *SPATA31C1*, *ZNF733P* and *OR2W5* pseudogenes were associated with patients’ survival. Due to pseudogene transcripts’ potential for cancer diagnosis, progression, and as therapeutic targets, our study can guide new research to HNC understanding and development of new target therapies.

## 1. Introduction

Head and neck cancer (HNC) is the eighth most common cancer worldwide, with more than 835,000 new cases and 431,000 deaths due to the disease per year [1]. HNC comprises tumors in the oral cavity, pharynx, and larynx, and nearly 95% of them are squamous cell carcinoma histological type tumors [2]. The classical risk factors for developing HNC are smoking habits and alcohol consumption [3]. The human papillomavirus (HPV), in particular HPV16, is detected in approximately 25% of HNC cases, especially in tumors located at the oropharynx, and serves as a favorable prognostic factor for these patients [4]. Most HNC patients are diagnosed with measurable locally advanced disease, and only about half of these patients achieve complete or partial responses after treatment [5]. Besides surgery, cisplatin (CDDP), combined or not with radiotherapy (RT), has been used in HNC patients’ treatment [6], but therapy resistance has been reported [7].

Genetic and epigenetic alterations also play an important role in HNC progression [8,9]. Single-nucleotide polymorphism, copy number variation, loss of heterozygosity, and post-transcriptional expression modulation (somatic or inherited) have already been described in HNC [8,9]. However, few or no biomarkers are currently used in clinical practice. Patients’ therapy resistance and poor survival rates highlight the need to find novel molecular biomarkers in HNC and investigate their potential mechanism in tumor initiation and progression.

## 2. Pseudogenes

In 1977, Jacq and his colleagues first established the term “pseudogene” to describe DNA sequences similar to a functional gene but with mutations (e.g., frameshifts, premature stop codon) that render their RNA or protein products nonfunctional [10,11]. The label of “junk DNA” fell apart many years ago, as research evidence indicating pseudogenes’ role in many biological processes has grown steadily [12]. Although most pseudogenes are too deteriorated to perform a biological function, it has been shown that at least 20% of them show transcriptional activity and that pseudogenic RNAs can be translated into proteins [13].

Pseudogenes can be categorized as unprocessed (unitary or duplicated) or processed [11]. The unprocessed pseudogenes feature a loss of productivity, expression of RNA or protein-coding ability, and are generated by point mutation, insertion, deletion, misplaced stop codon, or frameshift from the parental gene. The unitary unprocessed pseudogenes differ from the duplicated ones in that the former had an established function (no parental gene) rather than being a more recent copy of a functional gene that was disabled [11]. In addition, the duplicated pseudogenes maintain the intron–exon structure [11]. The processed pseudogenes are formed by integrating the parental gene reverse-transcribed mRNA transcript into the genome by retro-transposition [11]. The processed pseudogenes lack introns, 5’ promoter sequence, and have flanking direct repeats and a 3’ polyadenylation tag [11]. In general, processed pseudogenes found within introns are more frequently expressed than processed pseudogenes found in intergenic regions due to the transcriptional process of their host genes [14].

Many pseudogenes hold conserved mutations in different species, indicating a common descent or evolutionary ancestry [11]. Moreover, established pseudogenes are passed from generation to generation, and they may be partially duplicated to create a new pseudogene [11]. Although nucleotides within the pseudogenes are conserved to maintain the original genetic elements, some cases of polymorphic pseudogenes have been identified and associated with cancer, such as *E2F3P1* (E2F transcription factor 3 pseudogene 1) in liver cancer [15], *MYLKP1* (myosin light-chain kinase pseudogene 1) in colon cancer [16], and *GBAP1* (glucosylceramidase beta pseudogene 1) in gastric cancer [17].

## 3. Pseudogene Transcripts

The transcription of pseudogenes depends on their genomic location, and it could be processed mainly into antisense RNAs, endogenous small interfering (endo-si)RNAs, or micro (mi)RNA sponges [18], acting as negative or positive gene regulators [19]. Pseudogene antisense RNAs can bind directly into sense RNA from the parental gene and inhibit its translation and protein production [20]. Pseudogene-derived endo-siRNAs are produced by cleaved pseudogene sense or antisense transcripts of double-stranded RNA or by the inverted repeat region of pseudogenes transcribed into hairpin-shaped RNA and sliced by the ribonuclease Dicer [20]. They are then separated into single strands and incorporated into the RNA-induced silencing complex to bind and degrade target mRNAs, inhibiting protein production [20]. Pseudogene transcripts can also act as competing endogenous RNAs (ceRNA) [21]. They may share miRNA response elements with other genes, especially parental ones, and positively regulate their expression by competing for the same pool of miRNAs, acting as miRNAs sponges [20]. In this process, pseudogene transcripts inhibit miRNA-target gene binding, allowing gene expression and modulating biological processes and tumor progression [22,23]. In addition, pseudogene transcripts in sense orientation may compete for miRNAs, RNA-binding proteins, and translational machinery with the parental gene [18].

Pseudogene transcriptome can vary during physiological and pathological processes, such as cancer [19]. Pseudogene transcripts have been described in diverse types of human cancer, acting as tumor promotors, facilitating cancer development, or as tumor suppressors, inhibiting cancer progression [19]. Moreover, pseudogene transcripts have also shown a prognostic role in HNC, highlighting its importance in patient prognosis and treatment outcomes [24].

## 4. Pseudogene Transcripts in HNC

Although largely understudied, some pseudogene transcripts have been described in HNC. Here, we divided the literature information by tumor location (HNC in general with mixed patients, oral cancer, and laryngeal cancer studies) and provided a summary of pseudogenes associated with HNC described in the literature in Table 1. To the best of our knowledge, there is no study of pseudogene transcripts in the pharynx only. Additionally, it is possible that some real pseudogenes are named as genes in certain studies, meaning that we may have missed some information while searching the literature for pseudogenes only.

### 4.1. HNC in General

In HNC tumor cells, loss of *PTENP1* (phosphatase and tensin homolog pseudogene 1) pseudogene transcript modulated malignant behavior and worst survival of HNC patients, leading to tumor cell proliferation, colony formation, and migration, possibly by interacting with its parental gene *PTEN* (phosphatase and tensin homolog) [25].

In HNC tumor cells and cell lines (FaDu, Cal-27, SCC4, and SCC9), *FKBP9P1* (FKBP prolyl isomerase 9 pseudogene 1) pseudogene transcript abundance was found to correlate with advanced tumor stage and poor prognosis of patients by enhancing tumor cell proliferation, migration, and invasion, possibly by interacting with the PI3K/AKT signaling pathway [26].

Using RNA-seq data of HNC patients, Xing et al. [24] predicted a pseudogene signature, including *LILRP1* (leukocyte immunoglobulin-like receptor pseudogene 1), *RP6-191P20.5* (retinitis pigmentosa 6 191P20.5 pseudogene), *RPL29P19* (ribosomal protein L29 pseudogene 19), *TAS2R2P* (taste 2 receptor member 2 pseudogene), and *ZBTB45P1* (zinc finger and BTB domain-containing 45 pseudogene 1) pseudogenes, related with poor patients’ prognosis, possibly explained by a pseudogene transcript–parental gene interaction, increasing angiogenesis, tumor cell proliferation, and migration [24].

Based on The Cancer Genome Atlas (TCGA) data and the University of Alabama Cancer Database (UALCAN), Grzechowiak et al. [27] described that *PTTG3P* (pituitary tumor-transforming 3 pseudogene) pseudogene transcript abundance was correlated with HNC lower T-stage (T1 or T2), positive HPV16 status, and poor prognosis [27]. Gene Set Enrichment Analysis indicated that *PTTG3P* was correlated with genes involved in DNA repair, oxidative phosphorylation, and peroxisome pathways [27].

Several pseudogenes (AC010677.5, TCEB2P2, RPL37P2, PPIAP26, WTAPP1, UNGP3, UBA52P8, RP11-490K7.4, UBA52P6, EIF4HP2, AC114737.3, RP1-89D4.1, POLR2KP1, CD8BP, RP11-54C4.1, UNGP1, YWHAEP7, and NPM1P25) and their parental genes were involved in HPV16 viral infection in HNC patients, indicating their potential involvement in ribosomal activation control and increased protein synthesis during HPV16-alternated cell cycle [28].

### 4.2. Oral Cancer

In oral cancer cells, based on bioinformatics methods, *FTH1P3* (ferritin heavy-chain 1 pseudogene 3), *GTF2IRD2P1* (GTF2I repeat domain-containing 2 pseudogene 1), and *PDIA3P* (disulfide isomerase family A, member 3 pseudogene) pseudogene transcripts were associated with oral pathogenesis and metastasis by interacting with their targets, *MMP1* (matrix metallopeptidase 1), *MMP3* (matrix metallopeptidase 3), *MMP9* (matrix metallopeptidase 9), *PLAU* (plasminogen activator urokinase), and *IL8* (interleukin 8) genes, involved in tumor cell proliferation, migration, and metastasis [29].

*FTH1P3* pseudogene transcript was also associated with oral cancer risk and prognosis by becoming a miRNA sponge for miR-224-5p and thereby modulating the expression of the *FZD5* (frizzled class receptor 5) gene, facilitating cell proliferation and colony formation [30]. Moreover, the *FTH1P3* transcript was associated with advanced tumor stage and poor prognosis in oral cancer [31]. *FTH1P3* could promote proliferation, migration, and invasion of tumor cells, possibly by PI3K/Akt/GSK3β/Wnt/β-catenin signaling [31].

In addition, the *PTENP1* transcript was found to act as a ceRNA, protecting the parental gene *PTEN* from miR-21 binding and therefore inhibiting tumor cell proliferation and colony formation [32].

### 4.3. Laryngeal Cancer

In laryngeal cancer, *HMGA1P6* (high mobility group at-hook 1 pseudogene 6) and *HMGA1P7* (high mobility group at-hook 1 pseudogene 7) pseudogene transcripts were identified as possible miRNA sponges, allowing the increase of *HMGA2* expression (high mobility group at-hook 2) and other oncogenic genes involved in proliferation and cell cycle progression, such as *CCNB2* (cyclin B2) and *WNT* (proto-oncogene WNT) family member genes, and epithelial–mesenchymal transition, such as *SNAIL* (snail family transcriptional repressor) and *TWIST1* (twist family BHLH transcription factor 1) genes [33].

In a laryngeal cancer patient, the *HLA-A* (major histocompatibility complex class I A) processed pseudogene (*HLA-A**31012) was identified from retro-transposition of the parental gene within a clonal tumor cell [34]. The *HLA-A**31012 pseudogene transcript was restricted to tumor cells since it was not amplified in normal laryngeal tissue nor peripheral blood leucocytes [34]. This transcript may have contributed to laryngeal tumor progression by providing tumor cells with the immune system escape [34].

In addition, the *FTH1P3* pseudogene transcript was associated with advanced tumor stage and worst overall survival in laryngeal cancer by enhancing cell proliferation, migration, and invasion, and inhibiting cell apoptosis, although the exact mechanism was not elucidated [35]. Using clinical materials from TCGA, a five-gene signature predicting survival of laryngeal cancer patients was established by Zhang et al. [36], where the pseudogene *DPY19L2P1* was included. However, the exact mechanism was not investigated further [36].

Since pseudogene transcripts identified in HNC are largely unexplored in the literature, we performed an *in silico* analysis to predict potential biomarkers and their pathways for future validation with functional analysis.

**Table 1 genes-12-01254-t001:** Pseudogene transcript, pseudogene–gene interaction, tumor effect, clinical outcome, tumor localization, and human papillomavirus status in head and neck cancer described in the literature.

Pseudogene Transcript	Pseudogene–Gene Interaction	Tumor Effect	Clinical Outcome	Tumor Localization	Tumor HPV Status	Reference
*PTENP1*	*PTEN*	Facilitates the aggressiveness of tumor	Poor prognosis	HN	Not specified	[25]
*FKBP9P1*	*PI3K*/*AKT*	Facilitates the aggressiveness of tumor	Poor prognosis	HN	Not specified	[26]
*LILRP1*	*LILRB1*	Not specified	Poor prognosis	HN	Not specified	[24]
*RP6-191P20.5*	*VSIG*	Not specified	Poor prognosis	HN	Not specified	[24]
*RPL29P19*	*PMEPA1*	Not specified	Poor prognosis	HN	Not specified	[24]
*TAS2R2P*	*KLK5*	Not specified	Poor prognosis	HN	Not specified	[24]
*ZBTB45P1*	*HEATR1*	Not specified	Poor prognosis	HN	Not specified	[24]
*PTTG3P*	*PTTG1* and *PTTG2*	Facilitates the aggressiveness of tumor	Tumor development and progression	HN	Mixed	[27]
*AC010677.5*	*RPL23*	Facilitates HPV16 infection	Tumor development	HN	Mixed	[28]
*TCEB2P2*	*TCEB2*	Facilitates HPV16 infection	Tumor development	HN	Mixed	[28]
*RPL37P2*	*RPL37*	Facilitates HPV16 infection	Tumor development	HN	Mixed	[28]
*PPIAP26*	*PPIA*	Facilitates HPV16 infection	Tumor development	HN	Mixed	[28]
*WTAPP1*	*MMP1*	Facilitates HPV16 infection	Tumor development	HN	Mixed	[28]
*UNGP3*	*UNG*	Facilitates HPV16 infection	Tumor development	HN	Mixed	[28]
*UBA52P8*	*UBA52*	Facilitates HPV16 infection	Tumor development	HN	Mixed	[28]
*RP11-490K7.4*	*GTF2A2*	Facilitates HPV16 infection	Tumor development	HN	Mixed	[28]
*UBA52P6*	*UBA52*	Facilitates HPV16 infection	Tumor development	HN	Mixed	[28]
*EIF4HP2*	*EIF4H*	Facilitates HPV16 infection	Tumor development	HN	Mixed	[28]
*AC114737.3*	*FDPS*	Facilitates HPV16 infection	Tumor development	HN	Mixed	[28]
*RP1-89D4.1*	*RPS24*	Facilitates HPV16 infection	Tumor development	HN	Mixed	[28]
*POLR2KP1*	*POLR2K*	Facilitates HPV16 infection	Tumor development	HN	Mixed	[28]
*CD8BP*	*CD8B*	Facilitates HPV16 infection	Tumor development	HN	Mixed	[28]
*RP11-54C4.1*	*RPLP1*	Facilitates HPV16 infection	Tumor development	HN	Mixed	[28]
*UNGP1*	*UNG*	Facilitates HPV16 infection	Tumor development	HN	Mixed	[28]
*YWHAEP7*	*YWHAE*	Facilitates HPV16 infection	Tumor development	HN	Mixed	[28]
*NPM1P25*	*NPM1*	Facilitates HPV16 infection	Tumor development	HN	Mixed	[28]
*FTH1P3*	*MMP1, PLAU, MMP3* and *IL8*	Increased cell proliferation and migration	Tumor development and progression	Oral cavity	Not specified	[29]
*FTH1P3*	miR-224-5p (*FZD5*)	Increased cell proliferation	Tumor development and progression	Oral cavity	Not specified	[30]
*FTH1P3*	PI3K/Akt/GSK3β/Wnt/β-catenin	Increased cell proliferation and migration	Tumor development and progression	Oral cavity	Not specified	[31]
*GTF2IRD2P1*	*MMP1, PLAU, IL8* and *MMP9*	Increased cell proliferation and migration	Tumor development and progression	Oral cavity	Not specified	[29]
*PDIA3P*	*PLAU*	Increased cell proliferation and migration	Tumor development and progression	Oral cavity	Not specified	[29]
*PTENP1*	miR-21 (*PTEN*)	Increased cell proliferation	Tumor development	Oral cavity	Not specified	[32]
*HMGA1P6*	*HMGA2*	Facilitates the aggressiveness of tumor	Tumor development and progression	Larynx	Not specified	[33]
*HMGA1P7*	*HMGA2*	Facilitates the aggressiveness of tumor	Tumor development and progression	Larynx	Not specified	[33]
*HLA-A**31012	Not specified	Facilitates immune system escape	Tumor development	Larynx	Not specified	[34]
*FTH1P3*	Not specified	Increased cell proliferation and migration	Tumor development and progression	Larynx	Not specified	[35]
*DPY19L2P1*	Not specified	Not specified	Poor prognosis	Larynx	Not specified	[36]

HPV: human papillomavirus, HN: head and neck mixed tumors.

## 5. Materials and Methods

The *in silico* analysis consisted of acquiring and evaluating TCGA data from HNC patients to identify potential pseudogenes enrolled in tumor development. TCGA is a comprehensive public database for key genomic changes in various types of cancers, and we can obtain deregulated pseudogenes from the available transcriptome profiles.

Next, we constructed co-expression networks to identify pairs of genes (or pseudogenes) showing similar expression patterns across samples and performed gene ontology (GO) enrichment analysis to map the relationship between pseudogenes and other genes involved in the same biological process. A workflow of the methodology used in this study is shown in Figure 1.

### 5.1. TCGA Data Analysis

To identify potential pseudogene transcripts in HNC, we used the TCGA database (https://portal.gdc.cancer.gov; accessed on 21 May 2021) to download pseudogene transcript data on patients with HNC with squamous cell subtype and corresponding tumor location (oral cavity, oropharynx, hypopharynx, and larynx). We selected genomic information of patients with tumors located at the lip, gum, palate, floor of mouth, tonsil, base of tongue, oropharynx, nasopharynx, hypopharynx, and larynx (*n* = 546). Tumors in ill-defined sites and with complex mixed, stromal, adenomas, or adenocarcinomas subtypes were excluded. Next, we selected only those cases with primary tumors as sample type (*n* = 439). From those 439 HNC patients, we selected the cases with pseudogene information. We included only the transcript classification (biotypes) denominated, transcribed unprocessed pseudogene, processed pseudogene, unprocessed pseudogene, transcribed processed pseudogene, polymorphic pseudogene, immunoglobulin variable region pseudogene, and unitary pseudogene. From the 439 HNC patients, 220 did not present pseudogene expression information, which reduced our sample size to 219 HNC patients. From the 219 HNC patients selected, we also downloaded the somatic genomic alterations pseudogene data, such as single-nucleotide variation (SNV), deletions, and insertions, due to their potential of altering pseudogene transcription.

### 5.2. Co-Expression Networks and GO Enrichment Analysis

We built co-expression networks to better understand the relationship between pseudogenes and other genes in the genome. In a co-expression network, genes that have similar expression variation among samples are clustered in the same module, and they are generally thought to be involved in the same biological process [37]. Using RNAseq data from 213 HNC patients available in TGCA, we built a tumor co-expression network using the Weighted Correlation Network Analysis (WGCNA) R software package from gene expression values (log2-converted FPKM) [38]. It was not possible to obtain RNAseq data from six HNC patients. We kept only protein-coding genes from the human genome assembly reference GRCh38.p13 and the ten most deregulated pseudogenes of each HNC location (HNC in general, oral cavity, oropharynx, hypopharynx, and larynx). We then removed genes and pseudogenes not expressed in at least half of the samples and built the network with 19,406 protein-coding genes and 31 pseudogenes identified in the TCGA data analysis. Additional parameters used to build the network with WGCNA were a soft thresholding power of 8 and a minimum number of 30 genes allowed in a module [39]. The modules were then classified in colors.

To gain more insights about the possible functions of the pseudogenes described above, we performed a GO enrichment analysis for all gene modules containing at least one of most deregulated pseudogenes from HNC using the GOATOOLS Python library [40] (false discovery rate (FDR) adjusted *p-*values < 0.05) and subsequently summarized and visualized using the web tool REVIGO [41].

### 5.3. Survival Analysis

The Kaplan–Meier Plotter (https://kmplot.com/analysis/; accessed on 25 May 2021), an online database established from gene expression data and survival information of cancer patients from the TCGA database [42], was assessed to evaluate the prognostic value of the pseudogene transcripts identified in our study in a cohort of 500 HNC patients. Unfortunately, patients’ clinicopathological aspects were not available. A Kaplan–Meier survival plot, log-rank *p*-value, and confidence interval (CI) were directly determined and displayed by the database. Patients were split by the expression median. Relapse-free survival (RFS) consists of the date of diagnosis to the date of relapse or last follow-up. Overall survival (OS) consists of the date of diagnosis until the date of death, due to any cause, or last follow-up.

## 6. Results and Discussion

### 6.1. TCGA Data Analysis

Clinicopathological aspects of the selected 219 patients are presented in Appendix A. We identified 370 pseudogene transcripts associated with HNC, where *SPATA31D5P* (SPATA31 subfamily D member 5 pseudogene), *HERC2P3* (hect domain and RLD2 pseudogene 3), *SPATA31C2* (SPATA31 subfamily C member 2), *MAGEB6P1* (melanoma antigen family B6 pseudogene 1), *SLC25A51P1* (solute carrier family 25 member 51 pseudogene 1), *BAGE2* (B melanoma antigen family member 2), *DNM1P47* (DNM1 pseudogene 47), *SPATA31C1* (SPATA31 subfamily C member 1), *ZNF733P* (zinc finger protein 733 pseudogene), and *OR2W5* (olfactory receptor, family 2, subfamily W, member 5) were found to be the most deregulated in HNC. The ten most deregulated pseudogene transcripts, chromosome location, gene family function, and involvement in cancer can be found in Table 2. The complete list of results can be found in Appendix A.

When we stratified patients’ information by tumor location, we found a few differences in pseudogene transcripts’ pattern that could be explained by different cancer behaviors [43], but some of the identified pseudogenes were present in more than one tumor location.

In oral cancer (*n* = 62), *SPATA31D5P*, *NBPF25P* (neuroblastoma breakpoint family member 25 pseudogene), *HSP90AB2P* (heat shock protein 90 alpha class B member 2 pseudogene), *NXF4* (nuclear RNA export factor 4 pseudogene)*, FOLH1B* (folate hydrolase 1B), *DNM1P47*, *BNIP3P1* (BCL2/adenovirus E1B 19 interacting protein 3 pseudogene 1), *PKD1L2* (polycystin 1-like 2 pseudogene), *BAGE2*, and *ZNF658B* (zinc finger protein 658B, pseudogene) were found to be the 10 most deregulated of 154 pseudogene transcripts identified (Table 2 and Appendix A).

In oropharyngeal cancer (*n* = 51), *POTEA* (POTE ankyrin domain family, member A), *MROH5* (maestro heat-like repeat family member 5), *MSL3P1* (male-specific lethal 3 homolog pseudogene 1), *HLA-H* (major histocompatibility complex class I, H pseudogene), *TUBB8P7* (tubulin beta 8 class VIII pseudogene 7), *SLC7A5P2* (solute carrier family 7 member 5 pseudogene 2), *DPY19L2P1* (DPY19L2 pseudogene 1), *TSSC2* (tumor suppressing sub-transferable candidate 2 pseudogene), *SPATA31C2*, and *NXF4* were found to be the 10 most deregulated of 109 pseudogene transcripts identified (Table 2 and Appendix A).

In hypopharyngeal cancer (*n* = 8), *DPY19L2P3* (DPY19L2 pseudogene 3), *SPATA31D5P, GBA3* (glucosidase beta acid 3), *PLEKHM1P* (pleckstrin homology domain containing, family M member 1 pseudogene), *DPY19L2P1*, *MST1P2* (macrophage-stimulating 1 pseudogene 2), *RP11-44F14.1*, *ADAM21P1* (ADAM metallopeptidase domain 21 pseudogene 1), *MAGEB6P1*, and *OR12D2* (olfactory receptor family 12 subfamily D member 2) were found to be the 10 most deregulated of 21 pseudogene transcripts identified (Table 2 and Appendix A).

In laryngeal cancer (*n* = 98), *HERC2P3, SPATA31D5P, SPATA31C2, SLC25A51P1*, *MAGEB6P1*, *SPATA31C1*, *BAGE2*, *PNLIPRP2* (pancreatic lipase-related protein 2)*, ZNF733P* and *DNM1P47* were found to be the 10 most deregulated of 287 pseudogene transcripts identified (Table 2 and Appendix A).

Only one pseudogene identified in our analysis, *DPY19L2P1*, has already been associated with HNC. *DPY19L2P1* was considered an independent prognosis predictor of laryngeal cancer, where its higher expression was associated with the worst OS, although the exact mechanism was not explored [36]. In our analysis, *DPY19L2P1* was included in the ten most deregulated pseudogenes in oropharyngeal and hypopharyngeal cancers. However, it was identified only as the 54th most deregulated pseudogene in laryngeal cancer, and its prognostic value could not be evaluated. Other described pseudogenes have already been associated with carcinogenesis, although the exact mechanisms were not reported or detailed in most studies. *HERC2P3* upregulation was associated with gastric cancer cell growth and migration by interacting with the Akt signaling pathway [44]. *BAGE2* was related to the tumor-specific antigen profile [45]. *FOLH1B* was found to be involved in metallopeptidase activity, and its upregulation was associated with aggressiveness and metastasis in prostate cancer [46]. *BNIP3P1* upregulation was found in breast cancer brain metastases [47]. *PKD1L2* upregulation was associated with a good prognosis in breast cancer patients [48] and colorectal cancer risk and poor survival in obese patients [49]. *POTEA* upregulation was associated with increased colorectal cancer risk [50]. *MSL3P1* upregulation was considered a non-invasive biomarker of renal cell carcinoma [51]. *HLA-H* upregulation was associated with cervical [52] and lung [53] carcinomas. *GBA3* lower expression was associated with poor prognosis of hepatocellular carcinoma patients [54]. *MST1P2,* binding to miR-133b, affected the chemoresistance of bladder cancer cells to cisplatin-based therapy [55] and promoted cervical cancer progression [56]. *PNLIPRP2* lower expression was found in pancreatic ductal adenocarcinoma [57].

We also identified 993 somatic genetic alterations in the 370 pseudogene transcripts identified in HNC, and SNV was the most common type (96.8%), followed by deletions (1.9%) and insertions (1.3%). The genetic variations of the 31 most deregulated pseudogenes from HNC and its subtypes (*SPATA31D5P*, *HERC2P3*, *SPATA31C2, MAGEB6P1*, *SLC25A51P1*, *BAGE2*, *DNM1P47, SPATA31C1*, *ZNF733P*, *OR2W5, NBPF25P, NXF4*, *BNIP3P1, PKD1L2*, *ZNF658B, POTEA*, *MROH5*, *MSL3P1*, *HLA-H*, *TUBB8P7, SLC7A5P2*, *DPY19L2P1*, *TSSC2, DPY19L2P3*, *GBA3*, *PLEKHM1P, MST1P2*, *ADAM21P1, OR12D2, PNLIPRP2* and *HSP90AB2P*) are presented in Appendix A. The pseudogenes *SPATA31D5P*, *HERC2P3*, *MAGEB6P1*, *SPATA31C2*, *SPATA31C1*, *SLC25A51P1*, *BAGE2*, *HSP90AB2P*, *OR2W5* and *REG1P* were the most genetically altered by several SNVs in the HNC patients. The complete approach can be found in Appendix A. 

From these identified pseudogenes, only *BAGE2* was already studied for genomic mutation profile and its copy number variation may be associated with the Robertsonian Down syndrome [58]. Pseudogenes’ genetic alterations, their potential of interfering in pseudogene transcription, and carcinogenesis have not been explored in the literature yet.

**Table 2 genes-12-01254-t002:** Most deregulated pseudogene transcripts in head and neck cancer patients identified from The Cancer Genome Atlas (TCGA) database, chromosome location, gene family function, and studies in cancer.

Tumor Location and Pseudogene Transcript	Chromosome Location	Gene Family Function	Studies in Cancer	Reference
Head and neck (*n* = 219)				
*SPATA31D5P*	9q21.32	UV response and DNA repair	None	[59]
*HERC2P3*	15q11.1	Cell growth and migration	Gastric	[44]
*SPATA31C2*	9q22.1	UV response and DNA repair	None	[59]
*MAGEB6P1*	Xp21.3	Tumor-specific antigen	None	[60]
*SLC25A51P1*	6q12	Mitochondrial NAD^+^ transporter	None	[61]
*BAGE2*	21p11.2	Tumor-specific antigen	Lung, colon, and breast	[62]
*DNM1P47*	15q26.3	Mitochondrial division	None	[63]
*SPATA31C1*	9q22.1	UV response and DNA repair	None	[59]
*ZNF733P*	7q11.21	Transcription factor	None	[64]
*OR2W5*	1q44	Cellular signaling	None	[65]
Oral cavity (*n* = 62)				
*SPATA31D5P*	9q21.32	UV response and DNA repair	None	[59]
*NBPF25P*	1q21.1	Neuronal modulation	None	[66]
*HSP90AB2P*	4p15.33	Cell proteostasis	None	[67]
*NXF4*	Xq22.1	RNA export from nucleus	None	[68]
*FOLH1B*	11q14.3	Metallopeptidase activity	Prostate	[46]
*DNM1P47*	15q26.3	Mitochondrial division	None	[63]
*BNIP3P1*	14q12	Autophagy and apoptosis	Breast cancer brain metastases	[47]
*PKD1L2*	16q23.2	Transmembrane protein	Colorectal and breast	[48,49]
*BAGE2*	21p11.2	Tumor-specific antigen	Lung, colon, and breast	[62]
*ZNF658B*	9p12	Transcription factor	None	[69]
Oropharynx (*n* = 51)				
*POTEA*	8p11.1	Apoptosis	Colorectal	[50]
*MROH5*	8q24.3	Uncertain	None	
*MSL3P1*	2q37.1	Transcription regulation	Renal and gastric	[51,70]
*HLA-H*	6p22.1	Immune homeostasis	Cervical and lung	[52,53]
*TUBB8P7*	16q24.3	Oocyte maturation	None	[71]
*SLC7A5P2*	16p12.2	Amino acid transporter	None	[72]
*DPY19L2P1*	7p14.2	Transmembrane protein	Larynx	[36]
*TSSC2*	11p15.4	Tumor suppressor	None	[73]
*SPATA31C2*	9q22.1	UV response and DNA repair	None	[59]
*NXF4*	Xq22.1	RNA export from nucleus	None	[68]
Hypopharynx (*n* = 8)				
*DPY19L2P3*	7p14.3	Transmembrane protein	None	[36]
*SPATA31D5P*	9q21.32	UV response and DNA repair	None	[59]
*GBA3*	4p15.2	Glucosylceramide hydrolysis	Liver	[54]
*PLEKHM1P*	17q24.1	Autophagy	None	[74]
*DPY19L2P1*	7p14.2	Transmembrane protein	Larynx	[36]
*MST1P2*	1p36.13	Cell invasion and apoptosis	Bladder and cervical	[55,56]
*RP11-44F14.1*	16q12.2	Unknown	None	
*ADAM21P1*	14q24.2	Cell adhesion and proliferation	None	[75]
*MAGEB6P1*	Xp21.3	Tumor-specific antigen	None	[60]
*OR12D2*	6p22.1	Cellular signaling	None	[65]
Larynx (*n* = 98)				
*HERC2P3*	15q11.1	Cell growth and migration	Gastric	[44]
*SPATA31D5P*	9q21.32	UV response and DNA repair	None	[59]
*SPATA31C2*	9q22.1	UV response and DNA repair	None	[59]
*SLC25A51P1*	6q12	Mitochondrial NAD^+^ transporter	None	[61]
*MAGEB6P1*	Xp21.3	Tumor-specific antigen	None	[60]
*SPATA31C1*	9q22.1	UV response and DNA repair	None	[59]
*BAGE2*	21p11.2	Tumor-specific antigen	Lung, colon, and breast	[62]
*PNLIPRP2*	10q25.3	Lipase activity	Pancreas	[57]
*ZNF733P*	7q11.21	Transcription factor	None	[64]
*DNM1P47*	15q26.3	Mitochondrial division	None	[63]

*n*: number of patients, UV: ultraviolet radiation, NAD^+^: nicotinamide adenine dinucleotide.

### 6.2. Co-Expression Networks and GO Enrichment Analysis

We performed co-expression network and GO enrichment analyses to map the relationship between pseudogenes and other genes in the genome. After analyzing 19,406 protein-coding genes and 31 pseudogenes, our co-expression networks contained 14,379 genes organized in 36 modules, with an average of 399 genes per module (median: 140 genes; range: 35–3645 genes; Appendix A).

From the 31 most deregulated pseudogenes from HNC and its subtypes, 14 of them (*SPATA31D5P, SPATA31C2, DNM1P47, NBPF25P, HSP90AB2P, NXF4, ZNF658B, POTEA, MROH5, HLA-H, DPY19L2P1, DPY19L2P3, GBA3,* and *PNLIPRP2*) were present in our network across 8 modules (light yellow, red, dark magenta, orange, black, salmon, ivory, and dark olive green). Although the other pseudogenes of interest did not show a co-expression profile similar enough to other genes to be assigned to a module according to the parameters used to build this network, they are still important for further studies in HNC based on our previous TCGA data analysis.

We chose to highlight three modules (light yellow, red, and dark magenta) that contain genes well-known to be involved in head and neck carcinogenesis based on two important reviews [76,77]. However, the other five modules also presented relevant gene associations that should be evaluated in further studies.

The light-yellow module contained the greatest number of pseudogenes (*NBPF25P, HSP90AB2P*, *ZNF658B,* and *DPY19L2P3*). Interestingly, this module also contains 12 genes known to participate in head and neck carcinogenesis (*CASP8*, involved in apoptosis; *NOTCH2* and *NOTCH3*, involved in cell differentiation; *TRAF3*, involved in antiviral response; *RB1* and *HRAS*, involved in cell cycle control and proliferation; *PTEN*, involved in apoptosis and cell cycle control; *CUL3* and *NFE2L2*, involved in oxidative stress response; *EP300*, involved in chromatin remodeling, and the transcription factors *IGF1R* and *TP63*) [76,77], suggesting that these genes may directly or indirectly interact with the pseudogenes in this module and play a role in HNC (Figure 2A). In addition, the light-yellow module is enriched in GO terms related to tumor development, such as regulation of cell communication, cellular response to stress, and intracellular transport (Figure 2B). The potential relationship between the described pseudogenes and genes in this module has not been explored in the literature yet.

The red module contains the pseudogene *DNM1P47* and the tumor suppressor *TP53* gene, mainly involved in cell cycle control and apoptosis [76,77] (Figure 3A), and is enriched in GO terms associated with regulation of cell proliferation, response to stress, and cell death (Figure 3B). The potential relationship between the *DNM1P47* pseudogene and the *TP53* gene has not been explored in the literature yet.

The dark magenta module contains the known tumor genes *HLA-A* and *HLA-B*, involved in immune response [77], very closely associated with the *HLA-H* pseudogene (Figure 4A), and is also enriched in GO terms important to cancer biology, such as cell proliferation, regulation of the immune system, regulation of gene expression, and Wnt signaling pathway (Figure 4B). The relationship of the *HLA-H* pseudogene and *HLA-A* and *HLA-B* genes has been described in the literature, and even related to lung cancer in the Asian population [53]. The SNV rs12333226, capable of modulating *HLA-A* and *HLA-H* expression levels, was suggested to exert an effect on lung cancer through these two immune-related genes and the pseudogene [53].

The complete list of genes in light-yellow, red, and dark magenta modules can be found in Appendix A. The other five modules consist of an orange module containing *DPY19L2P1*, a black module containing *NXF4*, *MROH5*, and *SPATA31D5P,* a salmon module containing *PNLIPRP2,* an ivory module containing *SPATA31C2*, and a dark olive-green module containing *POTEA* and *GBA3* pseudogenes. The list of genes interacting with those pseudogenes and the enriched GO terms for these modules can be found in Appendix A.

### 6.3. Survival Analysis

To investigate the prognostic value of the pseudogenes identified by the TCGA database in HNC, we assessed the online database The Kaplan–Meier Plotter. In this analysis, only the prognostic value of the ten most deregulated pseudogenes identified in HNC general samples was assessed (*SPATA31D5P*, *HERC2P3*, *SPATA31C2*, *MAGEB6P1*, *SLC25A51P1*, *BAGE2*, *DNM1P47*, *SPATA31C1*, *ZNF733P* and *OR2W5*) because the online database does not allow to stratify the patients by tumor location.

We observed that lower expression of *SPATA31D5P, SPATA31C2, BAGE2, SPATA31C1, ZNF733P* and *OR2W5* pseudogene transcripts was associated with the worst RFS of HNC patients. Patients with lower expression presented 2.56, 2.63, 2.33, 3.57, 3.03, and 3.03 more chances respectively, to relapse, compared to other patients (Figure 5A).

In addition, we observed that higher expression of *SPATA31D5P*, *ZNF733P* and *OR2W5* pseudogene transcripts was associated with the worst OS of HNC patients. Patients with higher expression presented 1.39, 1.56, and 1.53 more chances respectively, of dying, compared to other patients. Moreover, lower expression of *SPATA31C2*, *BAGE2* and *SPATA31C1* pseudogene transcripts was associated with the worst OS of HNC patients. Patients with lower expression presented 1.43, 1.39, and 1.43 more chances respectively, of dying, compared to other patients (Figure 5B). *MAGEB6P1*, *SLC25A51P1* and *DNM1P47* pseudogene transcripts were not available in The Kaplan–Meier Plotter database, so we could not perform survival analysis. The *HERC2P3* pseudogene transcript did not influence RFS or OS in the HNC patients. However, in the literature, *HERC2P3* upregulation was associated with cell migration in gastric cancer, an independent prognosis factor [44]. The prognostic values of *SPATA31D5P*, *SPATA31C2*, *BAGE2*, *SPATA31C1, ZNF733P* and *OR2W5* have not been explored in the literature yet.

## 7. Conclusions

The biology of cancer is complex and not fully understood so far. Pseudogenes lost the label of “junk DNA” and are now known to be modulators of gene expression and potential biomarkers for cancer risk and prognosis [12]. Pseudogene transcripts regulate gene expression by directly binding at the target mRNA or by functioning as miRNA sponges, impeding miRNA–target mRNA binding [20]. Thus, pseudogene transcripts can function as negative or positive gene regulators [20]. 

In HNC, some pseudogene transcripts have been studied and associated with tumor aggressiveness, HPV16 infection, and prognosis [24,25,26,27,28]. However, their role in HNC remains poorly explored, while patients’ therapy resistance and poor survival rates highlight the need of finding novel molecular biomarkers. In our *in silico* analysis, we identified potential pseudogene transcripts, their genetic alterations, their interactions, and potential pathways in HNC progression and prognostics. Therefore, this study can guide new research to HNC understanding and development of new target therapies.

## Figures and Tables

**Figure 1 genes-12-01254-f001:**
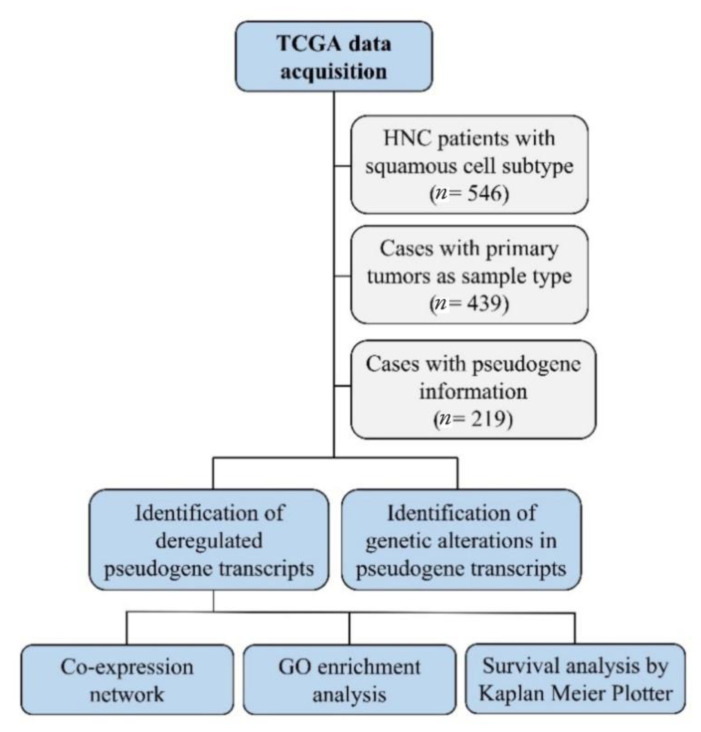
The workflow of the *in silico* analysis. This strategy detected deregulated pseudogene transcripts in head and neck cancer, their genetic alteration, gene interactions and pathways, and their role in patients’ survival.

**Figure 2 genes-12-01254-f002:**
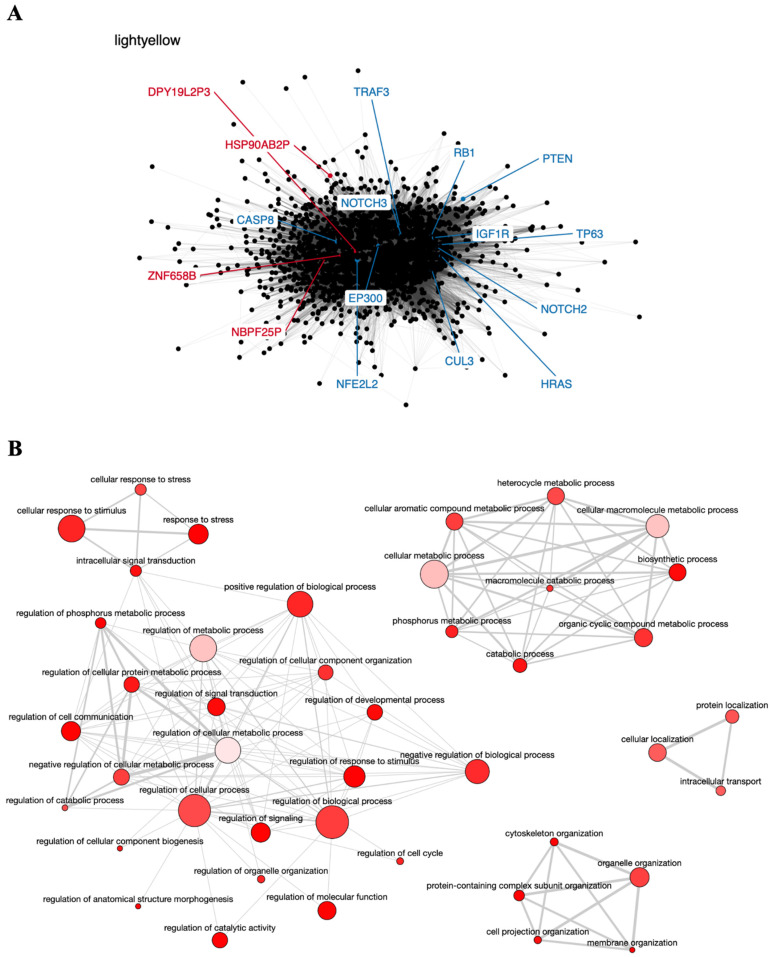
Co-expression network and GO enrichment analysis. Light-yellow module containing *NBPF25P, HSP90AB2P*, *ZNF658B* and *DPY19L2P3* pseudogenes from co-expression network of 213 head and neck cancer (HNC) patients and gene ontology (GO) enrichment analysis. (**A**) Network highlighting pseudogenes (red letter) and HNC-related genes (blue letter) constructed using the Weighted Correlation Network Analysis (WGCNA) R software package. (**B**) Enriched GO terms identified by GOATOOLS Python library and summarized by the web tool REVIGO. Circles colored with darker red indicate GO terms with lower *p*-values in our enrichment analysis, while the size of the circles indicates the frequency of a GO term in the GO annotation database. GO terms that are highly similar have thicker lines.

**Figure 3 genes-12-01254-f003:**
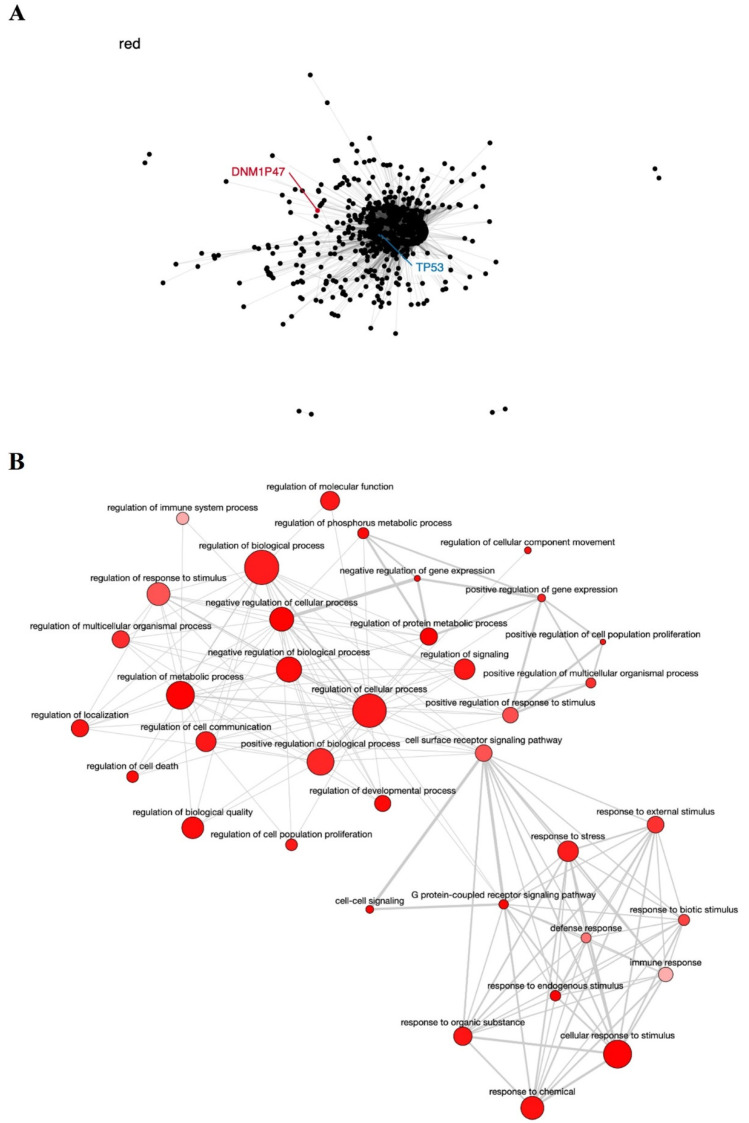
Co-expression network and GO enrichment analysis. Red module containing *DNM1P47* pseudogene from co-expression network of 213 head and neck cancer (HNC) patients and gene ontology (GO) enrichment analysis. (**A**) Network highlighting pseudogene (red letter) and HNC-related gene (blue letter) constructed using the Weighted Correlation Network Analysis (WGCNA) R software package. (**B**) Enriched GO terms identified by GOATOOLS Python library and summarized by the web tool REVIGO. Circles colored with darker red indicate GO terms with lower *p*-values in our enrichment analysis, while the size of the circles indicates the frequency of a GO term in the GO annotation database. GO terms that are highly similar have thicker lines.

**Figure 4 genes-12-01254-f004:**
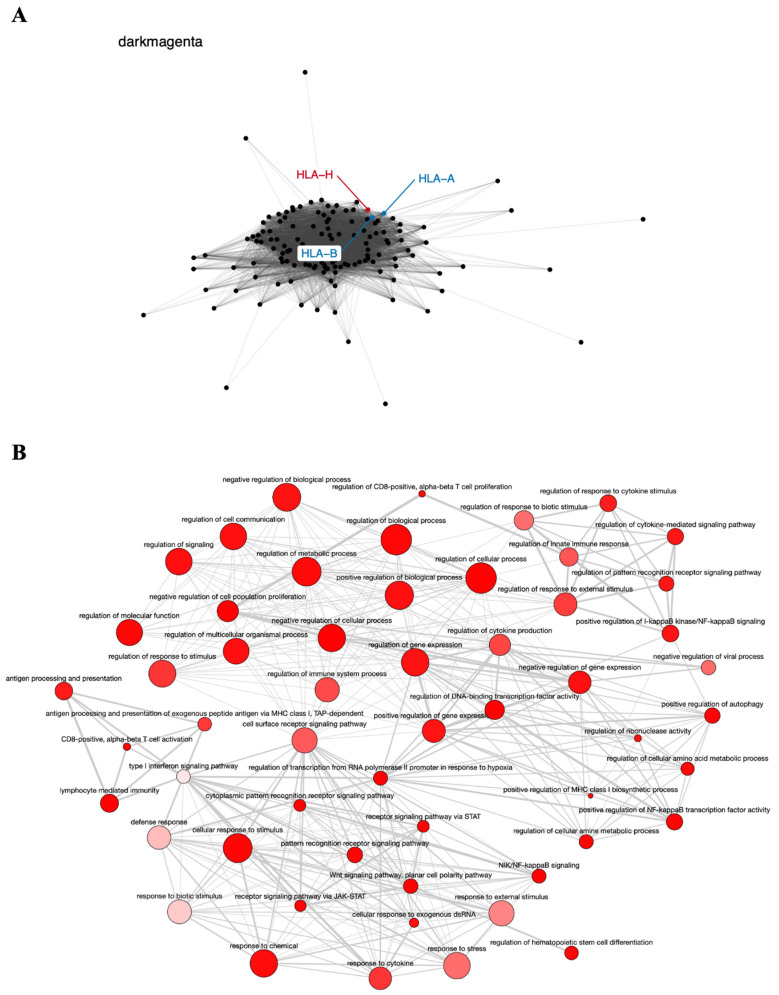
Co-expression network and GO enrichment analysis. Dark magenta module containing *HLA-H* pseudogene from co-expression network of 213 head and neck cancer (HNC) patients and gene ontology (GO) enrichment analysis. (**A**) Network highlighting pseudogene (red letter) and HNC-related genes (blue letter) constructed using the Weighted Correlation Network Analysis (WGCNA) R software package. (**B**) Enriched GO terms identified by GOATOOLS Python library and summarized by the web tool REVIGO. Circles colored with darker red indicate GO terms with lower *p*-values in our enrichment analysis, while the size of the circles indicates the frequency of a GO term in the GO annotation database. GO terms that are highly similar have thicker lines.

**Figure 5 genes-12-01254-f005:**
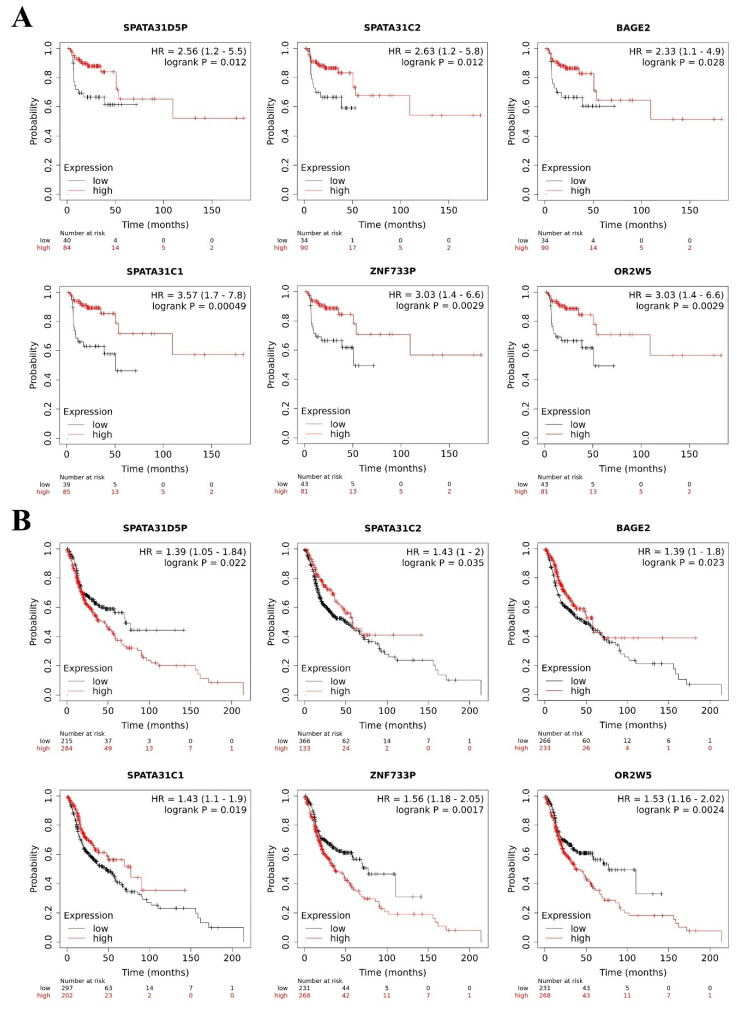
Pseudogene transcripts and head and neck cancer (HNC) patients’ survival. Prognostic value of deregulated pseudogene transcripts in HNC patients by Kaplan–Meier Plotter online database. (**A**) Lower expression of *SPATA31D5P*, *SPATA31C2*, *BAGE2, SPATA31C, ZNF733P* and *OR2W5* pseudogene transcripts indicated worst relapse-free survival in HNC patients. (**B**) Higher expression of *SPATA31D5P*, *ZNF733P*, and *OR2W5* pseudogene transcripts and lower expression of *SPATA31C2*, *BAGE2* and *SPATA31C1* pseudogene transcripts were associated with worst overall survival of HNC patients.

## Data Availability

The data presented in this study are available in the Appendix A.

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
