# Peer review of "Pseudogene Transcripts in Head and Neck Cancer: Literature Review and In Silico Analysis"

_genes, 2021, doi:10.3390/genes12081254_

Round 1

Reviewer 1 Report

The authors analyzed TCGA data of head and neck cancer (HNC) samples in order to investigate the role of pseudogene transcripts involved in HNC risk and they performed standard bioinformatics analysis such as WGCNA, functional enrichment, and survival analysis in order to reveal new putative prognostic biomarkers. The manuscript addresses a relevant topic to the scientific community and the study is scientifically sound, but some improvements and clarifications need to be performed to the presentation of the study. My detailed comments are given below.

Major

  • The abstract should be re-written in order to be more fluent. In the current version, it appeared as a list of performed analyses. An effective abstract should include: (1) a brief background of the question that avoids statements about how a process is not well understood; (2) a description of the approach and the results framed in the context of their conceptual interest; and (3) an indication of the broader significance of the work.
  • I would recommend a substantive revision of the Introduction in order to improve the state-of-the-art description. The authors missed citing important articles about ceRNAs, such as the first evidence of long non-coding acting as ceRNA (Poliseno et al Nature 2010) as well as other important examples of ceRNA mechanism in human diseases that are missing. See:
    1. Poliseno et al. "A coding-independent function of gene and pseudogene mRNAs regulates tumour biology", Nature, 2010
    2. Russo et al. "Interplay between long noncoding RNAs and MicroRNAs in cancer." Computational Cell Biology. Humana Press, New York, NY, 2018. 75-92
    3. Conte, et al. "An Overview of the Computational Models Dealing with the Regulatory ceRNA Mechanism and ceRNA Deregulation in Cancer." Pseudogenes (2021): 149-164.
  • It would be very useful for the reader to better following the performed analyses if the authors added a workflow of their analysis.
  • Each figure should have a title, in addition to the description.
  • Survival analysis:
  1. On my knowledge Kaplan Meier plotter allows to select TCGA datasets for performing the survival curves (pan-cancer analysis tab), including also the dataset of Head and Neck patients. Why did the authors choose to use GEO instead of using the TCGA dataset for the same patients that they analyzed?
  2. The authors should specify the criteria used for diving the patients in the two groups (median, upper-lower quartile, etc)
  3. Moreover the number of patients within the low and high expression group should be specified in the KM plots.
  • WGCNA analysis: It is not clear to me how the authors exploited the WGCNA analysis. More details should be provided. In particular:
  1. How many modules do they found?
  2. I know that WGCNA permits also to incorporate in the analysis external sample information (clinical/phenotypical traits) in order to screen for modules and intramodular hubs that relate to a trait of interest (e.g., disease status) and thus suggesting possible key roles of a specific network module in the phenotypic characterization. Did the authors use some traits of interest in their WGCNA analysis? If yes, Are there some modules with positive/negative correlations with some traits of interest? If not, why did the authors not integrate this information into the WGCNA analysis?
  3. How is the “gene significance” defined in this case? How did the authors select the hub genes? Did they choose a threshold on Module membership and/or gene significance? The authors should better clarify this point.

Minor

  • The overall writing quality could be improved. The authors should check the English language with a native speaker and check the entire paper avoiding careless errors, spelling mistakes, and fixing some wording.
  • First references seem not well-formatted, they are out of the document bounds.
  • Check all the commas. Add a comma before “and” whenever more than two elements have been listed.

Author Response

Response to Reviewer 1 Comments

Point 1: The abstract should be re-written in order to be more fluent. In the current version, it appeared as a list of performed analyses. An effective abstract should include: (1) a brief background of the question that avoids statements about how a process is not well understood; (2) a description of the approach and the results framed in the context of their conceptual interest; and (3) an indication of the broader significance of the work.

Response 1: We appreciate the reviewer suggestion, and we inform that the abstract was re-written to fulfil the above criteria, especially the addition of the main results of the study.

Point 2: I would recommend a substantive revision of the Introduction in order to improve the state-of-the-art description. The authors missed citing important articles about ceRNAs, such as the first evidence of long non-coding acting as ceRNA (Poliseno et al Nature 2010) as well as other important examples of ceRNA mechanism in human diseases that are missing. See:

  1. Poliseno et al. "A coding-independent function of gene and pseudogene mRNAs regulates tumour biology", Nature, 2010
  2. Russo et al. "Interplay between long noncoding RNAs and MicroRNAs in cancer." Computational Cell Biology. Humana Press, New York, NY, 2018. 75-92
  3. Conte, et al. "An Overview of the Computational Models Dealing with the Regulatory ceRNA Mechanism and ceRNA Deregulation in Cancer." Pseudogenes (2021): 149-164.

 Response 2: We appreciate the reviewer’s considerations, and we inform that we added the three articles mentioned above at the new version of the manuscript (page 3, Pseudogene transcripts section).

Point 3: It would be very useful for the reader to better following the performed analyses if the authors added a workflow of their analysis.

Response 3: We agree with the reviewer that the workflow helps in understanding the analysis, and we added one in page 8, Materials and Methods section.

Point 4: Each figure should have a title, in addition to the description.

Response 4: We inform the reviewer that we added a short tittle for each figure.

Point 5: Survival analysis: 1) On my knowledge Kaplan Meier plotter allows to select TCGA datasets for performing the survival curves (pan-cancer analysis tab), including also the dataset of Head and Neck patients. Why did the authors choose to use GEO instead of using the TCGA dataset for the same patients that they analyzed? 2) The authors should specify the criteria used for diving the patients in the two groups (median, upper-lower quartile, etc). 3) Moreover the number of patients within the low and high expression group should be specified in the KM plots.

Response 5: Indeed, the sentence in the original manuscript related to GEO dataset for Kaplan Meier plotter was wrong. We appreciate the reviewer’s comment, and we corrected this information in the new version of the manuscript. We have split patients by expression level median, and we also added this information in the new version of the manuscript (page 9, Material and Methods - Survival analysis). Moreover, Figure 4 was modified to present the number of patients in each group.

Point 6: WGCNA analysis: It is not clear to me how the authors exploited the WGCNA analysis. More details should be provided. In particular: 1) How many modules do they found? 2) I know that WGCNA permits also to incorporate in the analysis external sample information (clinical/phenotypical traits) in order to screen for modules and intramodular hubs that relate to a trait of interest (e.g., disease status) and thus suggesting possible key roles of a specific network module in the phenotypic characterization. Did the authors use some traits of interest in their WGCNA analysis? If yes, are there some modules with positive/negative correlations with some traits of interest? If not, why did the authors not integrate this information into the WGCNA analysis? 3) How is the “gene significance” defined in this case? How did the authors select the hub genes? Did they choose a threshold on Module membership and/or gene significance? The authors should better clarify this point.

Response 6: We appreciate for the reviewer comments, and we would like to explain that:

1) We found 36 modules and respective summary information is provided at the manuscript (page 14, Results and Discussion - Co-expression networks and GO enrichment analysis). More detailed information about the modules can be found in the Supplementary Dataset 2.

2) The reviewer raised an excellent point. We did not present any trait association with the gene modules for two reasons:

- The main goal for performing this co-expression network analysis was to better understand the relationship between promising pseudogenes involved in general head and neck cancer and other genes in the genome, particularly those already described in the literature to be associated in cancer-related processes, and then perform a gene ontology enrichment analysis on gene modules with these pseudogenes.

- Although we had a few interesting clinical traits with low missing data (tumor stage, location, age at diagnosis, and vital status), none of them showed a strong and statistically significant (FDR-adjusted p-value < 0.5) correlation with any of the modules identified in our WGCNA analysis. We felt that the trait-module correlation results would not add any relevant information to answer our main question described above and would possibly decrease the readability of our manuscript.

3) Based on the definitions provided by Langfelder and Horvath (2008) in the WGCNA paper, gene significance refers to the correlation between a gene and a trait. As explained above, since we did not perform any association analysis between gene modules and traits, the definition of gene significance is not applicable to our analysis. We also did not assign module membership (i.e., correlation of the module eigengene and the gene expression profile) to identify gene hubs. In our understanding, this metric is most relevant when you have no pre-defined genes of interest in your analysis and want to find which genes within a module correlated with a trait are “more important”. Such genes would then be considered for further analysis. In our case, the pseudogenes highlighted in the main text are our main genes of interest. We specifically wanted to know if these genes were associated to any cancer related genes (from known head and neck cancer genes described in the literature) and processes (from our gene ontology enrichment analysis).

We highlight that this approach is the first step for other studies that could confirm pseudogene transcripts’ role, their targets and pathways, and their possible to association with clinical pathological aspects in a larger cohort.

Point 7: The overall writing quality could be improved. The authors should check the English language with a native speaker and check the entire paper avoiding careless errors, spelling mistakes, and fixing some wording.

Response 7: We appreciate for the reviewer comment, and we would like to inform that the writing was checked for any errors or mistakes.

Point 8: First references seem not well-formatted, they are out of the document bounds.

Response 8: We inform the reviewer that the references were corrected.

Point 9: Check all the commas. Add a comma before “and” whenever more than two elements have been listed.

Response 9: We appreciate for the reviewer comment, and we would like to inform that all commas were revised.

Reviewer 2 Report

Carron J et. al firstly summarized the past studies on pseudogene transcripts in HNC specifically, followed by performed the bioinformatic study based on TCGA database. The manuscript was impressive and systematically in my view. However, some minor points should improve the manuscript.

  1. When the final 219 HNC patients were picked from the 439 samples, the authors should specify the selection criteria more clearly, and should describe the advantages of final selections.
  2. Lots of candidate pseudogene transcripts were identified through the bioinformatic studies using TCGA database, however only one, BAGE2 wea ever studied before. The authors should describe more clearly about this publication and should give the perspective of their own on others candidate genes in the conclusion or discussion part.
  3. It should be better if another database was also analyzed.

Author Response

Response to Reviewer 2 Comments

Point 1: When the final 219 HNC patients were picked from the 439 samples, the authors should specify the selection criteria more clearly, and should describe the advantages of final selections.

Response 1: We would like to inform the reviewer that from the 439 head and neck cancer patients identified at TCGA database, 220 did not present pseudogene expression information, which reduced our sample size to 219 patients. In the subsequent analyses, we only looked at samples with pseudogene information. We added this information in the new version of the manuscript (page 9, Methods - TCGA data analysis section).

Point 2: Lots of candidate pseudogene transcripts were identified through the bioinformatic studies using TCGA database, however only one, BAGE2 wea ever studied before. The authors should describe more clearly about this publication and should give the perspective of their own on others candidate genes in the conclusion or discussion part.

Response 2: Indeed, BAGE2 was the only one previously studied in the mutation profile aspect. BAGE2 copy number variation was associated with gene dosage change and breakpoint analysis in Robertsonian Down syndrome. However, the study did not thoroughly explore the exact consequences of this copy number variation.

We would like to comment that the only pseudogene identified in our analysis from TCGA database that was studied before in head and neck cancer is DPY19L2P1. DPY19L2P1 pseudogene was considered an independent prognosis predictor of laryngeal cancer, where its higher expression was associated with the worst overall survival. However, the exact pathway that DPY19L2P1 pseudogene could be involved was not explored in the study. We added this information in the new version of the manuscript (page 11, Results and Discussion - TCGA data analysis section).

Point 3: It should be better if another database was also analyzed.

Response 3: We agree with the reviewer that another database could also be used at the analysis. However, we choose to use only one database to keep samples information as homogeneous as possible. Besides, TCGA allows to download original sample information and easy sharing of cancer data than others.

Round 2

Reviewer 1 Report

The authors addressed all the issues raised in the first round of revisions, thus I suggest publishing the manuscript.